# The Role of microRNAs in NK Cell Development and Function

**DOI:** 10.3390/cells10082020

**Published:** 2021-08-07

**Authors:** Arash Nanbakhsh, Subramaniam Malarkannan

**Affiliations:** 1Laboratory of Molecular Immunology and Immunotherapy, Blood Research Institute, The Blood Center of Wisconsin, Milwaukee, WI 53226, USA; arash.nanbakhsh@bcw.edu; 2Department of Pediatrics, The Medical College of Wisconsin, Milwaukee, WI 53226, USA; 3Department of Microbiology and Immunology, The Medical College of Wisconsin, Milwaukee, WI 53226, USA; 4Department of Medicine, The Medical College of Wisconsin, Milwaukee, WI 53226, USA

**Keywords:** microRNAs, NK cells, Mirc11, inflammation

## Abstract

The clinical use of natural killer (NK) cells is at the forefront of cellular therapy. NK cells possess exceptional antitumor cytotoxic potentials and can generate significant levels of proinflammatory cytokines. Multiple genetic manipulations are being tested to augment the anti-tumor functions of NK cells. One such method involves identifying and altering microRNAs (miRNAs) that play essential roles in the development and effector functions of NK cells. Unique miRNAs can bind and inactivate mRNAs that code for cytotoxic proteins. MicroRNAs, such as the members of the *Mirc11* cistron, downmodulate ubiquitin ligases that are central to the activation of the obligatory transcription factors responsible for the production of inflammatory cytokines. These studies reveal potential opportunities to post-translationally enhance the effector functions of human NK cells while reducing unwanted outcomes. Here, we summarize the recent advances made on miRNAs in murine and human NK cells and their relevance to NK cell development and functions.

## 1. Introduction

In humans, peripheral blood NK cells represent 10–15% of the circulating lymphocytes [1].

NK cells do not express clonotypic receptors. In humans, NK cells are defined by the expression of CD56 (NCAM or Leu-19), and a subset of mature NK cells express the activating Fc receptor CD16 [2,3]. NK cells belong to innate lymphoid cells (ILCs) [4]. There are three distinct groups within the ILCs in humans and mice [4,5]. NK cells are a subset of group 1 ILCs and produce proinflammatory cytokines including IFN-γ, GM-CSF, and TNF and chemokines such as CCL3, CCL4, CCL5, XCL1, and XCL2 upon stimulation [6]. NK cells also mediate antitumor cytolytic functions [6].

Recent studies have elucidated many physiological functions of non-coding RNAs in multiple cell types [7,8]. Based on their functions, non-coding RNAs can be subgrouped into small nucleolar RNAs (snoRNAs), small nuclear RNAs (snRNAs), short interfering RNAs (siRNAs), and microRNAs (miRNAs) [9]. MicroRNAs are the most studied non-coding RNAs and are small (~22 nucleotides) sequence-specific guides that direct the Argonaut protein complex (Figure 1), an essential component of the RNA-induced silencing complex (RISC), to suppress protein translation by Watson–Crick pairing to mRNA transcripts [10,11]. Mature miRNAs target the four 3′ untranslated regions (3′-UTR) of mRNAs using a ‘seed sequence’ of 2–8 nucleotides. Once bound, miRNAs either inhibit translation or degrade the target transcript [12]. The miRNAs are highly conserved among different species [13]. The expression of miRNAs is cell-specific, and each cell type expresses a distinct set of miRNAs at various stages of maturation, differentiation, or activation [14].

In the last ten years, an increasing number of reports have demonstrated the role of miRNAs in controlling human and murine NK development, activation, and effector functions [15,16,17,18,19,20]. The mechanisms by which these miRNAs operate are complex and novel and provide detailed blueprints to formulate augmented and personalized therapies against aggressive malignancies. Moreover, recent findings also offer approaches to contain the production of inflammatory cytokines without dampening the cytotoxic potency of NK cells. Either introducing unique miRNAs or blocking others into NK cells can help alter the functions of target mRNAs. Here, we summarize the recent advances in identifying and characterizing miRNAs in murine and human NK cells and their relevance to clinical utilization. 

## 2. Role of miRNA in NK Cell Development

Natural killer (NK) cells develop and mature in the BM, secondary lymphoid tissues (SLTs), including tonsils, spleen, and LNs [21]. Human NK cells can be divided into transcriptomically distinct developmental and functional subsets [22]. More than 400 and 300 miRNAs have been identified in human and mouse NK cells, respectively [23,24]. Several of them play essential roles in NK cell development and maturation [25]. 

### 2.1. MicroRNA-Mediated Commitment to the NK Cell Lineage

Earlier studies showed *miRNA-181* regulates the development and maturation of mouse or human NK cells. Ectopic expression of *miR-181* into CD34^+^ hematopoietic progenitor cells (HPC) promoted the generation of human primary CD56^+^ NK cells in an in vitro culture system [26]. Overexpression of *miR-181a* and *miR-181b* increased the absolute numbers of B and T cells. In NK cells, these microRNAs are expressed in a developmental stage-specific manner [26]. A Nemo-like kinase (NLK), an inhibitor of Notch signaling, is a target of *miR-181* in NK cells, which is essential for NK cell development. NLK negatively modulates the transcription of Notch-dependent target genes by blocking the formation of a DNA-binding ternary complex consisting of Notch, CSL, and Mastermind (Figure 2A).

The second example is *miR-150*, which is highly expressed in murine and human lymphocytes, plays a role in B and T cell development [27]. Lack of *miR-150* resulted in a reduced number of mature NK cells in the peripheral tissues of *miR-150^−/−^* mice [28]. This lack of maturation was due to the overexpression of c-Myb, which is a target of *miR-150* and subsequently affects the expression of its downstream targets [28,29]. In contrast to its absence, overexpression of *miR-150* augmented NK cell maturation and trafficking. Expression of activation receptors (NKG2D, NCR1, CD244, Ly49D, Ly49H, and NK1.1) or inhibitory receptors (NKG2A, Ly49A, Ly49C/I, Ly49G2) was comparable between NK cells derived from *Mirc11^−/−^* and WT mice [20]. Functional phenotyping using the CD27, CD11b, and KLRG1 markers implied that the maturation of NK cells proceeds normally in the absence of *Mirc11*. Compared to *miRNA-181* [30] or *miR-150* and *miR-15/16* [28,31], *Mirc11* does not appear to silence transcripts whose products are obligatory for the commitment, survival, and maturation of developing NK cells [20]. One reason could be that although the members of the *Mirc11* cluster are present in naïve NK cells, their expression levels are comparatively low and are increased following activation of the NK cells [20]. 

MicroRNA *miR-29b* is highly expressed in NK cells, and it specifically downregulated transcription factors Eomes and Tbx21 in a murine model and thus played an essential role in both the conversion of NK progenitor to immature NK cells and in the terminal maturation and functions of NK cells [32,33,34]. These findings set the stage for future studies to utilize naturally occurring miRNAs that can help in in vitro generation of custom-made human NK cells for clinical utilization. The *Let-7* family consists of multiple members [35,36]. Members of the *Let-7* family are among the abundantly expressed microRNAs in NK cells [37]. In CD8^+^ T cells, target transcripts of *Let-7* include Myc and Eomes [38]. Similarly, miR-139 also targets Eomes in CD8^+^ T cells [39]. Identical mechanisms are predicted to be operative in NK cells during early development and future studies are warranted to validate this claim (Figure 2B).

### 2.2. MicroRNA-Mediated Control of NK Cell Maturation

The role of *miR-15a/16-1* was first identified in B cell development and malignancies [40]. The conditional deletion of *miR-15a/16-1* in NK cells significantly reduced NK cell maturation and proliferation. A 50% reduction in the expression of *miR-15/16* caused a significant decrease in the maturation of the CD27^−^CD11b^+^ (stage IV) murine NK cells [23,40]. The lack of *miR-15/16* reduced the levels of *Myb* mRNA and its protein expression in primary immature NK cells, demonstrating a positive role for these microRNAs. The resulting phenotype of lacking *miR-15/16* is similar to *miR-150* mice. Analogous to *miR-150*, *miR-15a/16-1* also targets *c-Myb* and modifies NK cell maturation and development [31]. However, these two families may employ opposite mechanisms to control the expression of c-Myb. MicroRNA *miR-142-5p/3p* plays a role in NK cell development and function [41,42]. Irrespective of having similar numbers of precursors of NK cells in the BM of *miR-142^−/−^* mice, their absolute number significantly reduced in the periphery, including spleen, blood, and lymph nodes. A global reduction of NK cells may be due to a defect in integrin expression resulting in reduced interaction within the BM niche essential for NK cell homeostasis. 

In humans, miRNAs *hsa-miR-31a-5p* and *hsa-miR-130a-5p* are exclusively expressed by CD56^bright^/CD16^−^ NK cells and *hsa-miR-181a-2-3p* is expressed by CD56^dim^ NK cells [19]. An intermediate expression of these miRNAs was found in CD56^bright^/CD16^dim^ NK cells, which strengthens the notion that CD56^bright^/CD16^−^ NK cells mature towards CD56^dim^ NK cells, and through an intermediate step represented by CD56^bright^/CD16^dim^ NK cells. MicroRNA *miR-146a* plays a role in human NK cell development [18,43,44]; it is a member of the *miR-146* family consisting of two evolutionarily preserved genes: *miR-146a* and *miR-146b*. This family plays a critical role in the inflammatory response of monocytes and lymphocytes through modification of multiple targets, including TNF receptor-associated factor 6 (TRAF6) and IL-1 receptor-associated kinase 1 (IRAK1), and play a role as a negative feedback regulator of the NF-κB pathway (Figure 2C). MicroRNA *miRNA-146a* also regulates the expression of killer Ig-like receptors (KIR) on NK cells [19]. The dual-luciferase reporter assay revealed that KIR2DL1 and KIR2DL2 are potential direct targets of *hsa-miR-146a-5p* [19]. MicroRNA miR-146a-5p possesses a complementary sequence that allows its binding to the 3′UTR of the KIR2DL1/KIR2DL2 mRNA and modulates its protein translation. These findings substantiate that miRNA might play a role in the licensing process of human NK cells during their development.

### 2.3. Additional Mechanisms Employed by miRNA during NK Cell Development

Each transcript can be targeted by multiple miRNAs. Similarly, a single miRNA can target multiple transcripts coding for distinct proteins. Transcripts possessing similar miRNA response elements (MREs) compete for the same miRNA. This process generates an additional level of post-transcriptional control of protein translation. These transcripts are known as competing-endogenous RNAs (ceRNAs) [45]. The existence of ceRNAs in NK cells has been recently established [46]. One of the long intergenic non-coding RNAs (lincRNAs), *EPHA6-1*, was induced by IFN-beta and secreted as part of exosomes from human lung carcinoma cell line A549 [46]. *Linc-EPHA6-1* functioned as a ceRNA for miRNA *miR-4485-5p*. This IFN-beta-induced competition for *miR-4485-5p* relieved the translational suppression on the transcript encoding NKp46 (NCR1). Augmented expression of NKp46 increased the cytotoxic potentials of human NK cells. Thus, ceRNAs provide a double-negative regulatory feedback loop in regulating receptors and proteins involved in NK cell activation.

In addition to this, functions of miRNAs can be regulated using exogenous factors such as ‘sponge’ RNA [47,48,49,50]. A sponge RNA can be designed to have multiple target sites for single or members of miRNA cistrons. Naturally occurring sponge RNAs have been proposed as a positive regulatory mechanism [48]. While these sponge RNAs are highly attractive as a therapeutic approach, their natural occurrence is debated. Synthetic sponge RNAs can specifically inhibit miRNAs with common seed sequence and augment the expression of target transcripts.

The ability of miRNAs to block translation can be affected by *cis*-regulatory elements present in the 3′ untranslated regions (3′-UTR) of transcripts. The majority of the miRNA-binding sites and regions where RNA-binding proteins interact are contained in 3′-UTR of transcripts between the stop codon and the poly(A) tail. Therefore, the 3′-UTRome plays a significant role in regulating the mRNA cleavage and polyadenylation which controls the translational efficacy, transcript stability/half-life, and subcellular localization. Since the majority of the miRNA-binding sites are present within the 3′-UTR, differential cleavage or polyadenylation can considerably impair the ability of miRNA targeting.

## 3. MicroRNA and NK Cell Effector Functions

NK cells mediate two distinct sets of effector functions. NK cells possess spontaneous lytic competence that does not require prior sensitization against cells undergoing a malignant transformation or are infected with a virus or other intracellular pathogens. The cytolytic function of NK cells can be initiated through several processes, including degranulation and death receptor ligation. It is critical for the clearance of diseased, infected, malignant, and dysfunctional cells. Besides this, NK cells are among the major sources of proinflammatory cytokines and chemokines. This ability of NK cells also let them help differentiate and polarize the adaptive immune responses. The role of miRNAs in regulating these processes is emerging.

MicroRNA *miR-146* negatively regulates IFN-γ production in IL-12/IL-18-activated human NK cells [44]. IRAK-1 and TRAF6 are the direct targets of *miR-146a*, and miRNA-146a significantly inhibited luciferase activity of the wild-type 3′-UTR of IRAK1 and TRAF6 in HeLa and HEK293T cells. Long exposure of primary human NK cells to IL-12 induces upregulation of specific microRNAs, including *miR-132*, *miR-212*, and *miR-200a* [51]. This induction is associated with gradual reduction of STAT4 levels and decreasing IFN-γ expression. After an initial burst of STAT4-mediated IFN-γ production within the first 16 h of IL-12 treatment, the effect diminished [51]. Therefore, the transcription of *miR-132*, *miR-212*, and *miR-200a* relates to the negative regulation of IL-12-mediated activation of NK cells. Inhibition of *miR-132*, *miR-212*, and *miR-200a* eliminates hyperresponsiveness in NK cells following prolonged IL-12 exposure. In contrast, overexpression of these microRNAs using pre-miRNAs mirrors hyperresponsiveness.

Although the expression of *miR-155* is minimal in non-stimulated mouse and human NK cells, it is increased significantly following stimulation with IL-12 and IL-18 [23,52,53,54]. Inositol phosphatase SHIP-1 is a direct target of *miR-155*, and *miR-155* acts as a positive regulator of IFN-γ production in human NK cells [53] (Figure 2D). Chronic mild stress causes upregulation of *miR-155* expression, which is associated with the augmented production of proinflammatory cytokines in rat NK cells via the ERK1/2 signaling pathway [54]. MicroRNA *miR-155* can also increase the expression of Noxa and SOCS1 in NK cells for enhanced survival and proliferation. Expression of *miR-155* is upregulated in NK cells following mouse cytomegalovirus (MCMV) infection [55]. Further in vivo studies are needed to elucidate the targets and signaling mechanisms mediated by *miR-155*. TGF-beta1 augments the expression of miR-27a-5p, which targets and downregulates the expression of CX_3_CR1 in NK cells leading to reduced trafficking [56]. MicroRNA miR-362 targets CYLD, a negative regulator of NF-κB signaling in NK cells, and thereby amplifies receptor-mediated activation and effector functions of NK cells [57]. Thus, overexpression of miR-362-5p promoted the production of IFN-γ, perforin, granzyme-B, and CD107a in CD56^+^ human primary NK cells. The link between miR-362-5p and CYLD was shown by silencing CYLD with a small interfering RNA (siRNA) that mirrored the effect of miR-362-5p overexpression [57]. These studies provide evidence that microRNAs can regulate both the cytotoxic potentials and the production of inflammatory cytokines from NK cells.

### 3.1. Mirc11 Cistron: A Divergent Role in Inflammation and Antitumor Cytotoxicity

For the following section, we took one miRNA cistron as an example and discussed the mechanism and clinical benefits of targeting it in NK cells. The *Mirc11* cistron encodes three independent miRNAs, *miR-23a*, *miR-24-2*, and *miR-27a*, on mouse chromosome #8 (#19 in humans) [58] (Figure 3A). Members of the *Mirc11* family are highly conserved across different species, including mouse, rat, horse, dog, and human genomes (Figure 3B).

*Mirc22* is a paralog of *Mirc11*, which encodes *miR-23b*, *miR-24-1*, and *miR-27b* located in mouse chromosome #13 (#9 in humans) [59]. Paralogs of *miR-23* and *miR-27* differ in one nucleotide, whereas the mature *miR-24-1* and *miR-24-2* are identical. Members of the *Mirc11* (*miR-23a*, *miR-24-2*) and *Mirc22* (*miR-23b*, *miR-24-1*) clusters are expressed at low but detectable levels as compared to abundantly expressed *miR-150*, *miR-29a*, *miR-16*, *miR-21*, *let-7a*, *let-7f*, *miR-24*, *miR-15b*, *miR-720*, *let-7g*, *miR-103*, and *miR-26a* in mouse and human NK cells [37,60]. The lack of the *Mirc11* cluster did not alter the development, maturation, or trafficking of NK cells. BM, spleen, lung, liver, and peripheral blood of *Mirc11^−/−^* mice contained mature NK cells in comparable numbers to WT [20]. Studies have shown that the *Mirc11* cluster is an essential positive regulator of inflammatory cytokine and chemokine production by targeting mRNA encoding ubiquitin modifiers. Members of the *Mirc11* cistron target and silence the translation of A20, Cbl-b, Cyld, and Itch. The lack of the *Mirc11* cluster led to increased translation of A20, Cbl-b, Cyld, and Itch, resulting in augmented deubiquitylation of K63 and ubiquitylation of K48 of TRAF6. Degradation of TRAF6 dampens the activation and nuclear translocation of NF-κB and AP-1 transcription factors and, therefore, the transcription of proinflammatory genes. The lack of the *Mirc11* cluster only moderately impaired the cytotoxic potentials of naïve or IL-15-cultured NK cells against EL4^H60^, RMA/S, B16F10, and YAC1 targets. Culturing NK cells that lacked the *Mirc11* cistron with IL-2 rescued the impairment in their cytotoxicity [20].

### 3.2. Mirc11 Cistron Has Only a Minimal Influence on NK Cell-Mediated Cytotoxicity

The *Mirc11* cluster did not enforce any regulation on the transcripts essential for the in vivo cytotoxic functions of NK cells. In vivo clearance of donor-derived splenocytes representing the ‘missing self’ (H-2^b^; β-2 microglobulin knockout C57BL/6 mice) or ‘non-self’ (H-2^d^; BALB/c mice) revealed that the lack of *Mirc11* impaired only the cytotoxicity of the ‘missing self’ targets. This may be due to the fact that one of the members of the *Mirc11* cistron, *miR-27a,* targets transcripts encoding perforin and granzyme B, and the absence of *miR-27a* augmented the cytotoxic potentials of human NK cells significantly [61]. Consistent with these findings, our recent work in profiling the transcriptomes of NK cells from either in vitro anti-NKG2D mAb-stimulated or from *Listeria monocytogenes*-challenged mice revealed partial reductions in the expression levels of perforin, granzyme B, granzyme A, granzyme C, and granzyme F in NK cells from *Mirc11^−/−^* mice [20]. Irrespectively of these in vivo observations, neither a significant reduction nor augmentation of NK cell-mediated cytotoxicity occurred in coculture experiments using the EL4-H60 model in which NK cells are activated through NKG2D. Collectively, these later findings suggest the reduced expression of perforin and various granzymes is insufficient to induce differences in cytotoxicity. These results also reveal that another aspect of the NK cell function is also disrupted. There was a fourfold increased expression of let-7a in *Mirc11*-deficient NK cells. Members of the let-7 family are known to negatively regulate effector functions of CTLs, and forced expression of let-7 targeted, degraded, and reduced the transcripts encoding effector molecules, including granzyme B. While these possibilities exist, the lack of *Mirc11* did not reduce the overall cytotoxic potentials of NK cells*^−/−^*.

### 3.3. Mirc11 Cistron Positively Regulates the Production of Inflammatory Cytokines in NK Cells

What are the molecular mechanisms by which *Mirc11* differentially regulates the cytotoxicity of NK cells? NK cell-derived IFN-γ is known to augment both activating (ICAM1) and inhibitory (MHC class I) ligands on target cells [62]. NK cells recognize target cells representing the ‘missing-self’ to employ their cell surface LFA-1 to interact with ICAM1 on target cells [63]. Thus, a significant reduction in the production of IFN-γ by NK cells from *Mirc11^−/−^* mice could exclusively affect the lysis of splenocytes from β-2 microglobulin knockout mice. Splenocytes from BALB/c mice express allo-MHC class I (H2-K^d^ and H2-D^d^) and H60, Rae-1, and MULT1 that are the ligands of the NKG2D receptor [64,65,66]. This could explain why the recognition and the in vivo clearance of BALB/c-derived splenocytes is not impaired in *Mirc11^−/−^* mice.

In support of this, coculturing of *Mirc11^−/−^* NK cells with EL4, EL4^H60^, RMA, RMA/S, YAC1, or B16F10 showed a significant reduction in IFN-γ production compared to WT mice. This defect in IFN-γ production could not be rescued by culturing naïve *Mirc11^−/−^* NK cells either with IL-2 or IL-15, which indicated that the impairment in cytokine production was due to a distinct alteration in the post-transcriptional mechanism. In addition, the production of other cytokines (TNF, GM-CSF) and chemokines (CCL3, CCL4, CCL5) by *Mirc11^−/−^* NK cells was also significantly reduced. This positive correlation of *Mirc11* and the generation of inflammatory cytokines is corroborated by the data collected in T cells by showing that the enforced overexpression of the individual members or the full cluster of *Mirc11* significantly elevated the production of IFN-γ, activation status (CD44^Hi^CD62L^Lo^), and cell proliferation in transgenic mice [67]. In addition, this overexpression of the *Mirc11* cluster resulted in the hyperactivation of T cells, differentially skewed the commitment of Th1, Th2, Th17, and induced T regulatory cells in a cytokine-dependent manner [67].

An earlier study also showed that both *Mirc11* and *Mirc22* were involved in containing Th2-mediated type 2 inflammation and lung pathology in an experimental mouse model of asthma [68]. The mechanism was primarily attributed to an IL-4-based network that is regulated by transcription factors Ikaros1 and Gata3 [68]. In another study, exposure of CD8^+^ T cells to tumor-derived TGF-β augmented the expression of the *Mirc11* cluster, which directly repressed the translation of BLIMP-1 [69]. These data demonstrate the regulatory role of Mirc11 extends beyond direct regulation of specific cytokines and chemokines and rather is required for maintaining a level of activation that enables the generation of multiple proinflammatory factors. Along these lines, our genome-wide transcriptomic analyses of anti-NKG2D mAb-stimulated *Mirc11^−/−^* NK cells further validate this role by demonstrating that loss of *Mirc11^−/−^* induces a broad inability to generate multiple inflammatory factors as opposed to specifically regulating individual cytokines and chemokines.

Importantly, the absence of the *Mirc11* cluster impaired the clearance of *L. monocytogenes* and considerably reduced the number of NK cells that produced IFN-γ. Recent studies have shown that two members of the *Mirc11* cluster, *miR-23a* and *miR-27a*, negatively regulate the expression of mitochondrial peptidylprolyl *cis*–*trans* isomerase (PPIF) in T cells during an established (14 days) *L. monocytogenes* infection [70]. By containing the expression levels of PPIF, *miR-23a* and *miR-27a* helped to maintain the mitochondrial integrity via restricting the influx of reactive oxygen species. This earlier study showed that the T cells lacking *miR-23a* and *miR-27a* were highly susceptible to TCR-mediated activation-induced cell death. Based on these data, we may expect *Mirc11^−/−^* NK cells to exhibit increased cell death during a *Listeria* infection. Additional work is warranted to compare the differential roles of the members of the *Mirc11* cluster between NK and T cells.

Uncontrolled inflammation forms the basis for allergy, asthma, and multiple autoimmune disorders [71]. Myriad signaling pathways have been implicated in initiating inflammatory responses. Among these, the NF-κB/Rel and AP-1 families of transcription factors play an indispensable role in the transcription of genes encoding inflammatory factors [72]. Transcriptomic signature of NK cells lacking *Mirc11* obtained either following anti-NKG2D-mediated activation (Figure 4) or an in vivo *L. monocytogenes* infection strongly predicted defects in the activation of NF-κB/Rel, AP-1, or both. A considerable decrease in the nuclear translocations of NF-κB/Rel and AP-1 in the NK cells lacking *Mirc11* corroborates the significant reduction in the transcript levels of genes that are positively regulated by these transcription factors. Reduction in the transcripts encoding *Myc*, *Irf4*, *Jun*, *Nfkbia*, *Cd69*, *Il2ra*, *Atf3*, and *Cd83* in NK cells from *Mirc11^−/−^* mice demonstrates a complete failure of NF-κB/Rel-mediated transcriptional regulation. Similarly, a considerable reduction in the transcript levels of *Atf4*, *Ccnd2*, *Dusp3*, *Nfe2l1*, *Stk40*, and *Ztbt32* that are direct targets of AP-1 confirmed a significant validated reduction in its transcriptional activity. These findings demonstrate that the *Mirc11* cluster is an active repressor of signaling pathways that inhibit the activation and nuclear translocation of NF-κB/Rel, AP-1.

Members of the *Mirc11* cistron *miR-23a*, *miR-24-2*, and *miR-27a* have the potentials to silence the translation of hundreds of mRNAs by targeting unique ‘seed’ sequences present in their 3′-UTR. Using TargetScan 7.1-based in silico analyses (http://www.targetscan.org, accessed on 28 July 2021) and in silico predictions, the potential target transcripts were identified in the total genome-wide RNA sequencing data from NK cells from *Listeria*-infected WT and *Mirc11^−/−^* mice based on ‘the aggregate probability of conserved targeting’ (P_CT_) [73]. Based on the ENCODE Data Coordination Center portal-based whole-genome alignments, we identified transcripts that were differentially expressed in NK cells from *Mirc11^−/−^* mice compared to WT mice [20]. These genomic data, along with additional biochemical analyses, indicated that the transcripts encoding A20, Cyld, Cbl-b, and Itch are the direct targets of the *Mirc11* cluster (Figure 5). The 3′-UTR of *Tnfaip3* contained one seed match for *miRNA-23a*-3p between 1664–1671 nts. Incorporation of this sequence into the 3′-UTR of the luciferase-encoding sequence significantly reduced its translation. The 3′-UTR of *Cblb* contained two seed matches at 3216–3236 and 5879–5901 nts targeted by *miR-27a*-3p and *miR-23a*-3p, respectively. Although the proximal seed match sequence for *miR-27a*-3p contained non-contiguous 13 nucleotides (out of 21) that were complimentary, we were unable to block the translation of luciferase. However, the distal sequence that was targeted by *miR-23a*-3p with a 7 mer seed match did contribute to translational repression, as indicated by the reduction in luciferase activity. The 3′-UTR of *Cyld* contained three sequences at 3842–3864, 4031–4051, and 5979–6000 nts, which were all targeted by *miR-24-2*-3p. Irrespective of the presence of three optimal seed matches, none of these were able to block the translation of luciferase. In this context, it is essential to note that *miR-24-1*, the paralog of *miR-24-2*, contains an identical sequence [59]. The 3′-UTR of *Itch* had two seed matches at 2882–2900 and 4649–4669 nts that were targeted by *miR-27a*-3p and *miR-23a*-3p, respectively. While no translational repression was seen from *miR-23a*-3p, incorporation of the target sequences in the 3′-UTR of luciferase indicated that *miR-27a*-3p could reduce the translation of luciferase.

Thus, the ability of the *Mirc11* cluster to directly target the transcripts encoding A20, Cyld, Cbl-b, and Itch provides a plausible mechanistic explanation for the reduction in the production of proinflammatory factors (Figure 6). Earlier work indicates a high expression level of miRNA-23a in human primary macrophages and its ability to target A20 to repress NF-κB activation and thereby reduce IL-6 and TNF corroborate the role of the *Mirc11* cistron in murine NK cells [74]. Deubiquitinating enzymes, such as A20 and Cyld, along with E3 ligases, including Itch and Cbl-b, are the central regulators of the TRAF6-NF-kB [75] and TRAF6-AP-1 pathways [76]. TRAF6, through its TRAF domain, promotes K63-linked autoubiquitination to function as a scaffold protein to recruit TAB1 and TAB2 [77]. TRAF6-mediated K63 polyubiquitination also recruits TAB2 and TAB3 to activate TAK1 [78], phosphorylating and activating IKKα and IKK, as well as nuclear translocation of the NF-κB complex [79]. A20 and Cyld are dual-function ubiquitin-editing enzymes that sequentially deubiquitinate K63- or Met-1 linkages and add K48-linked ubiquitin using their E3 ligase function [80,81]. Both A20 and Cyld can independently associate with Itch. Cyld interacts with Cbl-b to remove K63 polyubiquitin chains and add K48 polyubiquitination [82,83,84]. Although the protein levels of Cyld were augmented in the NK cells lacking *Mirc11*, there was no reduction in the activity when its seed match sequence was placed at the 3′-UTR of luciferase. The post-translational repression of both deubiquitinases and E3 ligases by *Mirc11* is a robust mechanism to dampen the inflammatory responses.

Thus, the *Mirc11* cistron offers a novel post-translational regulatory mechanism of inflammatory responses by microRNAs. The clinical relevance is highlighted by the evolutionarily conserved expression of the *Mirc11* cistron in human cells, including NK cells. Indeed, transduction of pri-miRNAs, an RNA hairpin with mature miRNA encoding either the individual or all three members of the *Mirc11* cistron, increased the production of IFN-γ in purified CD3ε^−^CD56^+^ human NK cells [20]. Future work is warranted to define the direct role of the *Mirc11* cluster as an underlying genetic susceptibility to inflammatory autoimmune diseases and malignancies [20].

## 4. Conclusions and Future Direction

Studies focusing on miRNA depending on the diverse aspects of NK cell biology are relatively new. More than a hundred microRNAs are expressed abundantly in resting NK cells, and there are others that can be induced or upregulated based on different types of stimulation, which makes it a necessity to have a better understanding of this post-transcriptional mechanism in NK cells development and function. MicroRNAs offer an exciting new opportunity to augment or contain distinct effector functions of lymphocytes, including NK cells. First, miRNAs can be employed as exceptional diagnostic biomarkers for unique malignancies and other infectious diseases. Secondly, miRNAs stand as a rheostat for gauging the efficacy of an ongoing immune response. Thirdly, miRNA offers complex mechanisms of biological processes and modes with which novel clinical therapies can be formulated. Cataloging the target mRNAs of specific miRNAs will provide molecular blueprints for drug identification and formulations. Following this, novel therapeutic approaches can be planned using synthetic sponge RNAs, AgomiR, and AntagomiRs. Synthetic sponge RNAs can be used to counter the negative influence of several miRNAs. This will help to augment the translation of specific mRNAs and thereby improve immune and cellular functions. In contrast, AgomiRs are double-stranded RNAs that act as endogenous miRNAs. AgomiRs can be used to downmodulate unwanted signaling pathways to augment a required immune function. AntagomiRs are synthetic 2-*O*-methyl RNA oligos that are designed to possess higher interacting potentials with the miRNA-associated gene-silencing complexes. AntogomiRs can exert a stronger effect on target transcripts. Formulations of these synthetic analogs open new avenues for both basic research and clinical therapy. Thus, future studies on miRNA hold promise for multiple diagnostic and therapeutic applications.

## Figures and Tables

**Figure 1 cells-10-02020-f001:**
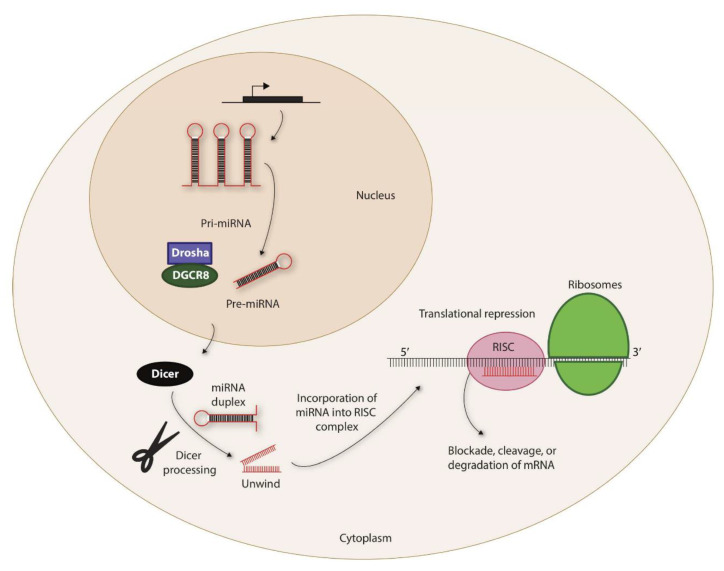
Biogenesis of miRNAs. The non-coding miRNAs are transcribed as Pri-miRNAs. This is processed by a complex containing Drosha and DiGeorge syndrome critical region 8 (DGCR8) within the nucleus to generate precursor-miRNA (Pre-miRNA). Pre-miRNAs are exported to the cytoplasm via exportin-5 that is located in the nuclear membrane. The 7-methylguanine-capped (m^7^G) pre-miRNAs depend on Dicer to complete their cytoplasmic maturation. The miRNA-induced silencing complex (RISC) recruits miRNA and the target transcripts to induce specific translational repression.

**Figure 2 cells-10-02020-f002:**
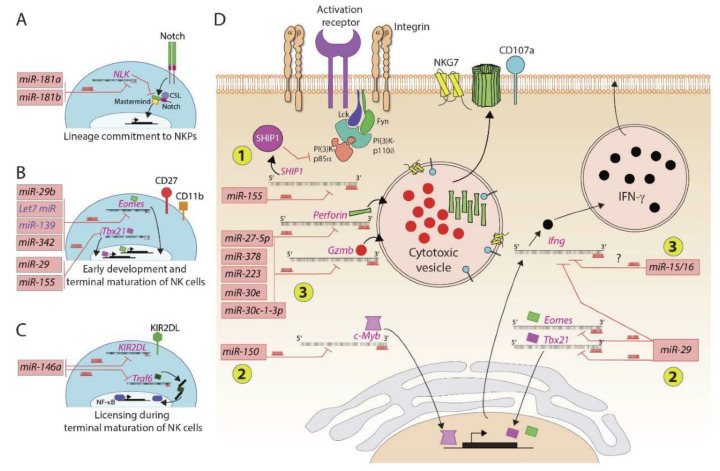
MicroRNAs regulate murine and human NK cell development and functions. Several microRNAs play an essential role in positively or negatively regulating the NK cell ontogeny. (**A**) Lineage commitment of progenitors to NKPs is regulated by *miR-181a* and *miR-181b* that target the transcript encoding NLK, a negative regulator of Notch signaling. (**B**) Early development and terminal maturation of NK cells are regulated by several miRNAs. The primary target of these miRNAs is the transcripts of transcription factors, including Eomes and Tbx21. (**C**) MicroRNA *miR-146a* targets transcripts encoding KIR2DL and TRAF6. KIRs interact with self MHC Class I and thereby determine the threshold of activation and licensing during the terminal maturation of NK cells. (**D**) MicroRNAs regulate the effector functions of NK cells. These regulatory effects occur at three levels: (1) *miR-155* degrades the transcript encoding lipid phosphatase SHIP that converts PI(3,4,5)P_3_ into PI(3,4)P_2_; conversely, an increase in the concentration of PIP_3_ augments the overall activation of NK cells; (2) miR-29 and miR-150 target messages encoding transcription factors, including Eomes, Tbx21, and c-Myb; (3) several microRNAs target the transcripts encoding effector molecules perforin, granzyme-b, and interferon gamma.

**Figure 3 cells-10-02020-f003:**
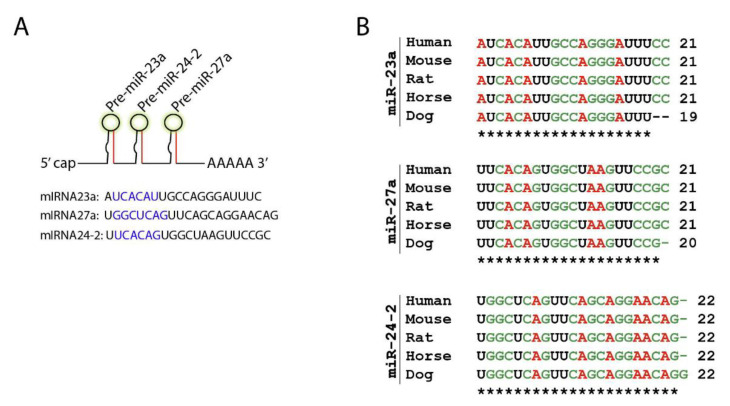
*Mirc11* family is evolutionarily conserved. (**A**) *Mirc11* cistron (*miRNA-23a* cluster) is a tri-miRNA cluster. It consists of three members, *miR-23a*, *miR-27a*, and *miR-24-2*, which are derived from a single primary mRNA transcript. The highlighted region forms the ‘seed sequences’ of these microRNAs. (**B**) *Mirc11* cistron is highly conserved in mouse, rat, horse, dog, and human genomes.

**Figure 4 cells-10-02020-f004:**
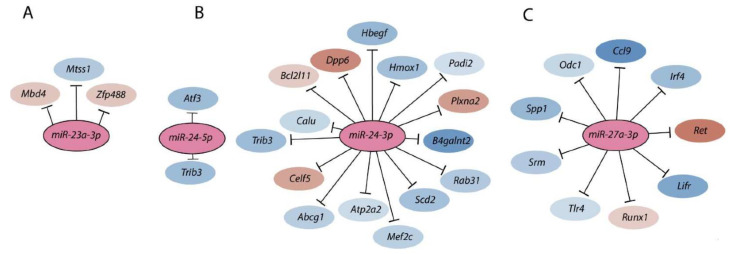
Members of the *Mirc11* family target multiple transcripts. Members of the *Mirc11* cistron *miR-23a* (**A**), *miR-24-2* (**B**), and *miR-27a* (**C**) have the potentials to silence the translation of hundreds of mRNAs by targeting unique ‘seed’ sequences present in their 3′-UTR. Using TargetScan 7.1-based in silico analyses (http://www.targetscan.org, accessed on 28 July 2021), potential target transcripts in the total genome-wide RNA sequencing data were identified from NK cells based on ‘the aggregate probability of conserved targeting’ (P_CT_). The ENCODE Data Coordination Center portal-based whole-genome alignments were used to determine the transcripts that are the targets of *Mirc11* family members. A select few target transcripts of the Mirc11 cistron are shown.

**Figure 5 cells-10-02020-f005:**
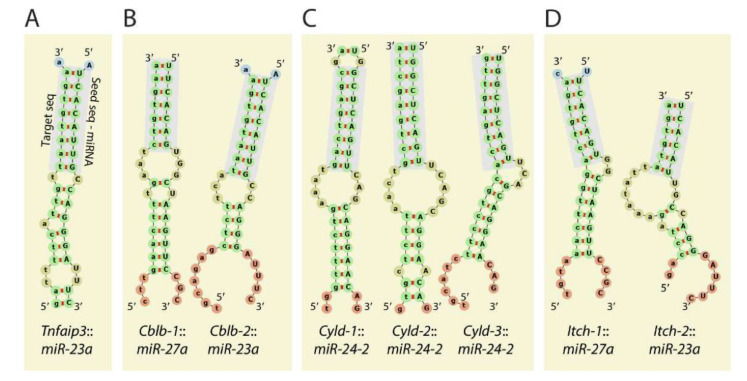
*Mirc11* targets members of E3 ligases. The 3′-UTR of the transcripts encoding E3 ligases A20 (TNFAIP3), Cblb, Itch, Cyld contains target sequences for the members of the *Mirc11* cluster. (**A**) The 3′UTR of *Tnfaip3* contained one seed match for *miRNA23a*-3p at 1664–1671 nts. (**B**) The 3′-UTR of *Cblb* contained two seed matches at 3216–3236 and 5879–5901 nts targeted by *miR-27a*-3p and *miR-23a*-3p, respectively. However, the role of *miR-27a*-3p and *miR-23a*-3p in degrading the *Cblb* transcript has not been established. (**C**) The 3′-UTR of *Cyld* contained three sequences at 3842–3864, 4031–4051, and 5979–6000 nts, which were all targeted by *miR-24-2*-3p. (**D**) The 3′-UTR of *Itch* contained two seed matches at 2882–2900 and 4649–4669 nts that were targeted by *miR-27a*-3p and *miR-23a*-3p, respectively.

**Figure 6 cells-10-02020-f006:**
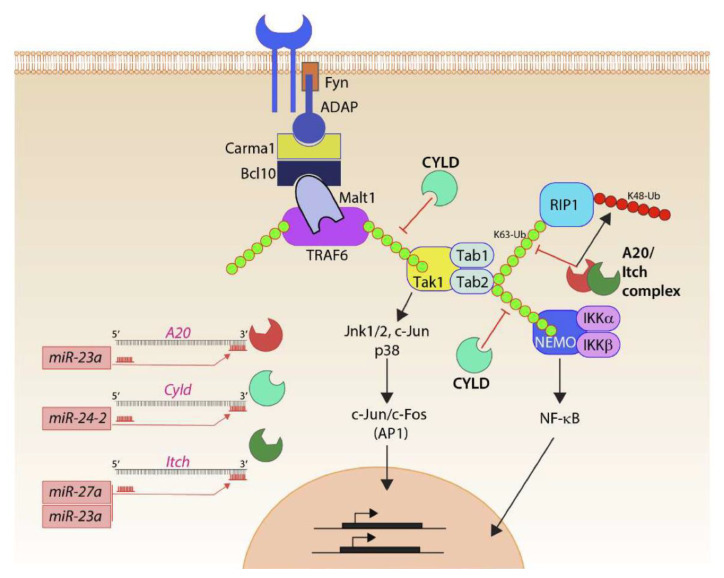
*Mirc11* and inflammation. Members of the *Mirc11* family function as cytoplasmic checkpoints to downregulate the functions of ubiquitin modifiers and augment translation of the transcripts encoding inflammatory cytokines. Signaling from activation receptors propagate by recruiting membrane-proximal Src kinase Fyn and scaffold protein ADAP. Functions of the ADAP include the assembly of the Carma1-Bcl10-Malt1 (CBM) signalosome. TRAF6 is one of the major ubiquitin E3 ligases in lymphocytes and auto-ubiquitylates K63 moieties of ubiquitin. K63-linked polyubiquitin functions as a scaffold in recruiting downstream effectors, including Tak1 involved in activating AP1 via Jnk1/2, p38, c-Jun. Tak1 also recruits Tab1, Tab2, and Tab3. K63 ubiquitylation of Tab2 allows the recruitment of Rip1 or Nemo and the eventual activation of NF-κB. Ubiquitin modifiers (Itch) and deacetylases (A20, Cyld) are involved in negatively regulating the functions of TRAF6 and Tabs. They achieve this by actively removing the K63 ubiquitin moieties and by adding K48-linked ubiquitin that marks proteins for proteasome-based degradation. *Mirc11* family members target Cyld, A20, and Itch. Using ‘seed sequences’, members of the Mirc11 family target sequences in the 3′ end of the transcripts; these microRNAs can function as the positive regulators of NF-κB and AP1 transcription factors.

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
