# Peer review of "The Role of microRNAs in NK Cell Development and Function"

_cells, 2021, doi:10.3390/cells10082020_

Round 1

Reviewer 1 Report

This is an extensive and scholarly review of the role of micro RNAs in regulating anti-tumour functions of NK cells, which is illustrated by five colourful and eye catching and detailed figures and I enjoyed reading it. I only have minor comments.

Minor comments

The title needs changing to something like: ‘The role of microRNAs in NK cell development and function’.

Line 23 – change ‘In humans peripheral blood, NK cells represent 10-15%..’ to ‘In humans, peripheral blood NK cells represent 10-15%...’

Line 65 – change to something like ‘2.1. miRNA-mediated commitment to the NK cell lineage’

Line 118  – change to something like ‘2.2. miRNA-mediated control of NK cell maturation’.

In my view sections 2.2 and 2.3 should be combined since they both represent the role of miRNA in aspects of NK cell maturation and so there is no need to separate these concepts.

TNF-alpha should be changed to ‘TNF’ to be consistent with the latest nomenclature.

Line 325 onwards – tab formatting issue.

Line 410 changes to ‘4. Conclusions and future direction’

Page 5 Line 197 onwards to apge 11, subheading ‘3.1. Mirc11 cistron: Divergent role in inflammation and anti-tumor cytotoxicity’. This section is quite large and complex biology is described. Is there any way of breaking this section up e.g. into the different strategies used by miRNA to regulate NK cell functions in inflammation and anti-tumour cytotoxicity?

The manuscript should be checked carefully for typos/grammar.

Author Response

Reviewer #1:

This is an extensive and scholarly review of the role of micro RNAs in regulating anti-tumour functions of NK cells, which is illustrated by five colourful and eye catching and detailed figures and I enjoyed reading it. I only have minor comments.

 Minor comments

 The title needs changing to something like: ‘The role of microRNAs in NK cell development and function’.

 --Thank you, and we have changed it.

Line 23 – change ‘In humans peripheral blood, NK cells represent 10-15%..’ to ‘In humans, peripheral blood NK cells represent 10-15%...’

 -- Thank you, and we have changed it.

Line 65 – change to something like ‘2.1. miRNA-mediated commitment to the NK cell lineage’

 -- Thank you, and we have changed it.

Line 118  – change to something like ‘2.2. miRNA-mediated control of NK cell maturation’.

 -- Thank you, and we have changed it.

In my view sections 2.2 and 2.3 should be combined since they both represent the role of miRNA in aspects of NK cell maturation and so there is no need to separate these concepts.

 -- Thank you, and we have changed it.

TNF-alpha should be changed to ‘TNF’ to be consistent with the latest nomenclature.

 -- Thank you, and we have changed it.

Line 325 onwards – tab formatting issue.

 --We are not sure what this is about. Hopefully, the journal staff will resolve it.

Line 410 changes to ‘4. Conclusions and future direction’

 -- Thank you, and we have changed it.

Page 5 Line 197 onwards to apge 11, subheading ‘3.1. Mirc11 cistron: Divergent role in inflammation and anti-tumor cytotoxicity’. This section is quite large and complex biology is described. Is there any way of breaking this section up e.g. into the different strategies used by miRNA to regulate NK cell functions in inflammation and anti-tumour cytotoxicity?

 --Thank you, and we have added two more subtitles in Section 3. Hope this helps the readers.

Reviewer 2 Report

This is a revision on the role of miRNAs in the differentiation and function of NK cells, with the aim of contributing to their clinical use by silencing the expression of mRNAs coding for toxic proteins while maintaining their effector functions. The authors review a number of miRNAs known to be involved in the commitment, survival, and maturation of developing NK cells. This is interesting because of the potential role of NK cells in anti-tumor theraphy.  Nevertheless, in its present form, this manuscript is just an enumeration of miRNAs and potential targets and lacks the insight for being published in a high-impact journal as “Cells” . For instance, Figure 3 shows a number of targets of the Mirc11 family. For an average reader this is almost useless, and should be complemented by a pathway analysis (GO Enrichment analysis, protein-protein interaction networks…) of these and other potential targets.

With the aim to improve the manuscript I would ask the authors to include new information in topics closely related to the function of miRNAs and adapted to the NK context such as:

1.- a new section on ceRNAs and other miRNA “sponges” active during the differentiation of NK cells.

2.- a new section on the variations in the 3’UTRome of NK-genic target mRNAs (alternative polyadenylation, alternative splicing) and its impact on miRNA binding and function. Changes in the 3’UTRome change the entire pattern of miRNA binding.

3.- a new section on the current use of miRNA agomirs and antagomirs in the context of the clinical use of NK cells. At the very end this is the key topic and it hasn’t been treated in the review

Author Response

Reviewer #2:

The manuscript should be checked carefully for typos/grammar.

 --Thank you, and we have carefully checked the text and have added corrections.

This is a revision on the role of miRNAs in the differentiation and function of NK cells, with the aim of contributing to their clinical use by silencing the expression of mRNAs coding for toxic proteins while maintaining their effector functions. The authors review a number of miRNAs known to be involved in the commitment, survival, and maturation of developing NK cells. This is interesting because of the potential role of NK cells in anti-tumor theraphy.  Nevertheless, in its present form, this manuscript is just an enumeration of miRNAs and potential targets and lacks the insight for being published in a high-impact journal as “Cells” . For instance, Figure 3 shows a number of targets of the Mirc11 family. For an average reader this is almost useless, and should be complemented by a pathway analysis (GO Enrichment analysis, protein-protein interaction networks…) of these and other potential targets.

With the aim to improve the manuscript I would ask the authors to include new information in topics closely related to the function of miRNAs and adapted to the NK context such as:

1. a new section on ceRNAs and other miRNA “sponges” active during the differentiation of NK cells.

--This information is added in the current version of the manuscript

2. a new section on the variations in the 3’UTRome of NK-genic target mRNAs (alternative polyadenylation, alternative splicing) and its impact on miRNA binding and function. Changes in the 3’UTRome change the entire pattern of miRNA binding.

--This information is added in the current version of the manuscript

3. a new section on the current use of miRNA agomirs and antagomirs in the context of the clinical use of NK cells. At the very end this is the key topic and it hasn’t been treated in the review

--This information is added in the current version of the manuscript

Reviewer 3 Report

The Authors report recent advances on the role of miRNAs in NK cell development, maturation and function.

The topic is very interesting and well defined, although in some parts it could be improved.

In particular, a representative figure on the miRNA synthesis and maturation processes could be introduced, in order to help the text understanding.

Some sentences should be revised in structure and construction. In some parts the Authors cite “our findings…” however, they report data of other Authors.

It would be interesting to introduce a paragraph concerning a possible prognostic value of miRNAs in the development of inflammatory processes in various pathological conditions.

Author Response

Reviewer #3:

The Authors report recent advances on the role of miRNAs in NK cell development, maturation and function. The topic is very interesting and well defined, although in some parts it could be improved.

--Thank you.

In particular, a representative figure on the miRNA synthesis and maturation processes could be introduced, in order to help the text understanding.

--Thank you, and we have included this in Figure 1.

Some sentences should be revised in structure and construction. In some parts the Authors cite “our findings…” however, they report data of other Authors.

--We apologize and have fixed this issue.

It would be interesting to introduce a paragraph concerning a possible prognostic value of miRNAs in the development of inflammatory processes in various pathological conditions.

--Thank you, and we have included this in the current version of this manuscript.

Round 2

Reviewer 2 Report

Authors have successfully addressed my concerns. This is a nice revision on an important topic